# FocalCodec: Low-Bitrate Speech Coding via Focal Modulation Networks

**Luca Della Libera**[1,2*] **Francesco Paissan**[3,2,4] **Cem Subakan**[5,1,2] **Mirco Ravanelli**[1,2]

[1]Concordia University  [2]Mila-Quebec AI Institute  [3]Fondazione Bruno Kessler
[4]University of Trento  [5]Université Laval

## Abstract

Large language models have revolutionized natural language processing through self-supervised pretraining on massive datasets. Inspired by this success, researchers have explored adapting these methods to speech by discretizing continuous audio into tokens using neural audio codecs. However, existing approaches face limitations, including high bitrates, the loss of either semantic or acoustic information, and the reliance on multi-codebook designs when trying to capture both, which increases architectural complexity for downstream tasks. To address these challenges, we introduce FocalCodec, an efficient low-bitrate codec based on focal modulation that utilizes a single binary codebook to compress speech between 0.16 and 0.65 kbps. FocalCodec delivers competitive performance in speech resynthesis and voice conversion at lower bitrates than the current state-of-the-art, while effectively handling multilingual speech and noisy environments. Evaluation on downstream tasks shows that FocalCodec successfully preserves sufficient semantic and acoustic information, while also being well-suited for generative modeling. Demo samples and code are available at https://lucadellalib.github.io/focalcodec-web/.

## 1 Introduction

Recent advancements in large language models [46, 10, 26, 17] have led to significant progress in natural language processing, enabling breakthroughs in tasks such as summarization, translation, question answering, code generation, and retrieval. Building on this success, the research community has extended these methods to other modalities, with speech emerging as a major area of interest. The impressive performance of text-conditioned audio and speech generation models [6, 11, 33, 69, 30], along with recent speech language models [85, 22, 16, 45], highlights the potential of token-based approaches for speech processing.

A key component of these pipelines is the *neural audio codec*, which compresses speech into tokens that downstream models can process. These tokens must preserve acoustic and semantic information to ensure effective representations for downstream tasks while maintaining high reconstruction quality. Another important requirement is a low token rate. As sequence length increases, capturing long-term dependencies becomes more challenging, and computational costs increase.

Despite recent progress, current codecs still face several challenges. Acoustic codecs [14, 34, 25, 76] achieve high-quality reconstruction but often rely on multiple codebooks, adding complexity to the design of downstream models. Additionally, they typically lack strong semantic representations. Hybrid codecs [86, 37, 16, 48] aim to combine both acoustic and semantic information while maintaining high-quality resynthesis. Still, they often depend on complex multi-codebook designs, explicit disentanglement, distillation losses, or supervised fine-tuning. Single-codebook designs [36,

---

*Correspondence to: `luca.dellalibera@mail.concordia.ca`

39th Conference on Neural Information Processing Systems (NeurIPS 2025).

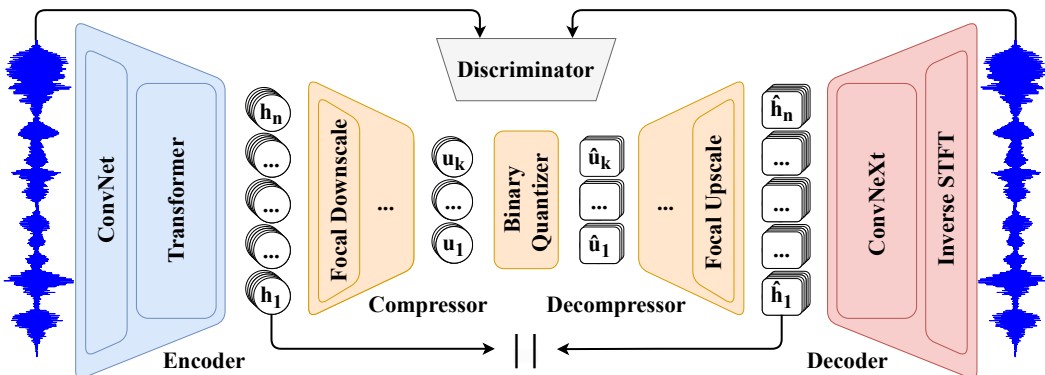

Figure 1: FocalCodec architecture. The encoder extracts features containing both acoustic and semantic information. These features are then mapped to a low-dimensional space by the compressor, binary quantized, and projected back by the decompressor. The decoder resynthesizes the waveform from these features.

21, 25, 76, 74] offer a simpler architecture but struggle to balance compression while maintaining both reconstruction quality and effective representations for downstream tasks, especially at low bitrates. To address these limitations, we introduce FocalCodec, an efficient low-bitrate codec based on focal modulation [81] that compresses speech into the space of a single binary codebook. FocalCodec achieves competitive performance in reconstruction at lower bitrates than the current state-of-the-art under a variety of conditions while also preserving sufficient semantic and acoustic information for downstream tasks.

Our contributions are as follows:

- We introduce FocalCodec, a novel **hybrid codec** featuring a compressor-quantizer-decompressor architecture that compresses speech using a **single binary codebook at ultra-low bitrates** (0.16 to 0.65 kbps).

- We propose a **focal modulation-based architecture** with strong inductive biases for speech, offering an efficient and scalable solution for tokenization.

- We demonstrate the versatility of FocalCodec through **comprehensive evaluations of reconstruction quality and performance in downstream tasks**, highlighting its potential for both discriminative and generative speech modeling.

Demo samples and code are available at https://lucadellalib.github.io/focalcodec-web/.

## 2   Related Work

**Acoustic Codecs.**   Acoustic codecs, built on the VQ-VAE [67] framework, aim for high-fidelity reconstruction. Notable advancements include hierarchical RVQ [83], lightweight architectures [14], improved RVQ techniques [34], and efficiency-driven designs [78, 57, 1]. Recent methods explore scalar quantization [41, 79], Mel-spectrogram discretization [4], and novel paradigms like diffusion- and flow-based decoding [75, 80, 49]. To reduce bitrate without compromising performance, multi-scale RVQ [63, 52] achieves improved compression by varying frame rates in deeper quantizers. However, its hierarchical design adds complexity to downstream applications, as it requires flattening the token sequences. Single-codebook designs [36, 21, 25, 76, 74] have emerged as a simpler, efficient alternative, delivering robust performance at low bitrates. Our codec aligns with this trend, leveraging a novel focal modulation architecture and a pretrained self-supervised encoder to efficiently unify semantic and acoustic representation learning.

**Semantic Codecs.**   Semantic codecs leverage self-supervised features from large models trained with contrastive [3] or predictive [23, 9] objectives and k-means clustering [39] for quantization, either from a single layer [50, 71] or multiple layers [44, 61]. Improvements upon this paradigm include replacing k-means with RVQ [24, 90, 20, 70], noise-aware [42] and speaker-invariant tokenization [8]. While these approaches effectively capture linguistic and content-related information, they often discard much of the acoustic detail, resulting in low speaker fidelity when a vocoder is trained

to resynthesize speech directly from these representations. To improve reconstruction quality, [90, 20, 70] incorporate continuous embeddings to capture prosody and speaker traits. However, this defeats the purpose of using speech tokenization for unified semantic and acoustic modeling. In contrast, our codec adopts a self-supervised architecture similar to semantic codecs but preserves both semantic content and acoustic detail through its compressor-quantizer-decompressor design and decoupled training strategy, ensuring high-quality reconstruction while preserving the advantages of semantic representations.

**Hybrid Codecs.** Hybrid codecs combine semantic and acoustic features to balance reconstruction quality and content representation. Some methods [28, 27, 88] employ multiple codebooks to disentangle speech into distinct subspaces, such as content, prosody, and timbre, while others [37] utilize dual encoders to separately capture content and fine-grained acoustic information. Semantic distillation [86, 16] has also been explored to enrich the first RVQ codebook with semantic information from HuBERT [23] and WavLM [9]. More recently, Parker et al. [48] trained a large-scale transformer-based VQ-VAE, achieving exceptional reconstruction quality at ultra-low bitrates. To enhance semantic content, they employed supervised fine-tuning on force-aligned phoneme data. Our codec also belongs to this category but instead of relying on complex multi-codebook designs with explicit disentanglement, distillation losses, or supervised fine-tuning, it is purely based on self-supervised learning. It compresses both semantic and acoustic information into a single codebook, pushing the boundaries of hybrid codec design at low bitrates.

## 3 FocalCodec

### 3.1 Architecture

The proposed codec is largely based on the VQ-VAE framework but incorporates **compressor** and **decompressor** modules between the encoder and decoder (see Figure 1). The discriminator is used only during training and is discarded afterward.

**Encoder.** To build a hybrid codec with a simple design, without relying on distillation losses or multiple encoders, the encoder must capture both acoustic and semantic information. This ensures high-quality reconstructions and expressive tokens for training downstream models. Self-supervised models like HuBERT and WavLM retain significant acoustic information in their lower layers [9], making them suitable for hybrid codecs. For instance, Baas et al. [2] show that a high-quality vocoder can be trained using continuous representations from layer-6 of WavLM-large. Following this approach, we use the **first 6 layers of WavLM-large**[2] as our encoder. However, effective quantization is critical for approximating continuous representations with sufficient granularity. Standard k-means clustering typically fails to preserve essential acoustic details [68]. To address this, we introduce a compressor-quantizer-decompressor design based on focal modulation, which allows for granular quantization that preserves both semantic and acoustic information.

**Compressor.** The compressor maps the encoder representations to a compact, low-dimensional latent space. Optionally, it can perform temporal downsampling to further reduce the frame rate. Prior work typically relies on convolutional, recurrent, or transformer-based architectures for compression. In contrast, we introduce a novel **focal downscaling** module, which combines a downscaling operation with a focal block. The downscaling step applies a linear projection to compress the feature dimension, while a 1D convolution can be used instead to additionally downsample along the time dimensions. To better capture periodic patterns, we follow [34] and apply Snake activations [89] after the projection.

To build a focal block, we replace the self-attention mechanism in the standard transformer block with focal modulation. Focal modulation [81] is an efficient alternative to self-attention that enables fine-to-coarse modeling and introduces useful inductive biases such as translation equivariance, explicit input dependency, time and channel specificity, and decoupled feature granularity. While originally designed for image and video processing, these properties also benefit speech modeling [15]. Unlike self-attention, which directly computes token-wise interactions, focal modulation first aggregates the global context and then modulates local interactions based on this aggregated representation. Intuitively, self-attention mixes tokens by first computing pairwise similarities and then aggregating, which can make the result sensitive to a few high-scoring neighbors. Focal modulation inverts this

---

[2]https://github.com/microsoft/unilm/tree/master/wavlm

order: it first forms a compact, multi-scale summary of the input (local + global context) and then uses this summary to modulate each token. This ensures that interactions are guided by the overall context rather than being dominated by individual tokens, while avoiding quadratic cost.

Formally, focal modulation computes output representation $\mathbf{y}_i$ for each input feature $\mathbf{x}_i$ in sequence $\mathbf{x}_{1:n}$ as:

$$\mathbf{y}_i = q(\mathbf{x}_i) \odot h \left( \sum_{\ell=1}^{L+1} \mathbf{z}_i^\ell \odot \mathbf{g}_i^\ell \right) \tag{1}$$

where $q(\cdot)$ and $h(\cdot)$ are linear projections, and $\mathbf{z}_i^\ell \in \mathbf{z}_{1:n}^\ell$ and $\mathbf{g}_i^\ell \in \mathbf{g}_{1:n}^\ell$ are the context and gating vectors at position $i$ and focal level $\ell \in \{1, \ldots, L+1\}$, with $\odot$ denoting element-wise multiplication. The context sequence $\mathbf{z}_{1:n}$ is obtained via a stack of depth-wise convolutions with increasing kernel sizes to capture dependencies from short to long range, with average pooling applied to the last level feature map to incorporate global information. Then, for each focal level, a point-wise convolution is used to compute the gating sequence $\mathbf{g}_{1:n}$. This hierarchical approach, operating at multiple granularities, makes focal modulation well-suited for processing speech features, enabling efficient and scalable representation learning in linear time while preserving long-range dependencies.

**Quantizer.** FocalCodec maps latent representations from the compressor into the codebook space of a **single quantizer**, eliminating the need for hierarchical designs in downstream models. To achieve this, while maintaining both reconstruction quality and efficiency, the quantizer should satisfy the following requirements: 1) given that the original waveform is already significantly compressed into a short sequence of latents, the quantizer must compensate by using a sufficiently large codebook size to reduce the quantization error; 2) the quantizer should make efficient use of the codebook capacity, avoiding under-utilization; 3) code lookup must remain efficient, despite the increased codebook size, to ensure fast inference.

To address these challenges, we employ **binary spherical quantization** (BSQ) [87], originally introduced for compression of images and videos. To the best of our knowledge, this is the first successful application of binary quantization in the speech domain. BSQ belongs to the category of lookup-free quantization (LFQ) methods [41, 82], i.e. it utilizes an implicit codebook, defined as:

$$\mathcal{C} = \left\{ -\frac{1}{\sqrt{L}}, \frac{1}{\sqrt{L}} \right\}^L, \tag{2}$$

which represents an $L$-dimensional hypercube projected onto a unit hypersphere. The codebook size is determined by the latent representation dimension $L$ as $|\mathcal{C}| = 2^L$. For example, latent representations of dimension 13 correspond to a codebook size of 8192. The quantization process consists of two steps. First, the input vector $\mathbf{v}$ of dimension $L$ is normalized to lie on the unit hypersphere:

$$\mathbf{u} = \frac{\mathbf{v}}{\|\mathbf{v}\|_2}. \tag{3}$$

Second, binary quantization with a normalization factor of $\sqrt{L}$ is applied independently to each dimension of $\mathbf{u}$:

$$\hat{\mathbf{u}} = \frac{\text{sign}(\mathbf{u})}{\sqrt{L}}, \tag{4}$$

where $\text{sign}(\cdot)$ denotes the sign function, with $\text{sign}(0)$ remapped to 1 to ensure the output always lies on the hypersphere. To make the quantization differentiable, we use the straight-through estimator [5]. BSQ offers several advantages over traditional quantization methods. First, the parameter-free implicit codebook is lightweight and computationally efficient. Second, empirical evidence [87] shows that the binary quantization bottleneck encourages high codebook utilization, even for large values of $L$, outperforming other lookup-free methods such as finite scalar quantization (FSQ) [41]. Third, the quantization error is bounded, resulting in faster convergence compared to vanilla LFQ, which does not normalize the representations. Finally, tying the codebook size to the latent dimension helps prevent performance degradation in downstream generative models when using larger codebooks [82].

**Decompressor.** The decompressor reconstructs the encoder continuous representations from the quantizer output. It closely mirrors the structure of the compressor, with the downscaling layers replaced by upscaling layers.

**Decoder.** Most codecs use symmetric architectures, where the decoder mirrors the encoder. However, some works [4, 25, 37] explore asymmetric designs with larger decoders to improve reconstruction quality. In this work, we adopt an asymmetric design but prioritize the encoder, allocating $\sim 5\text{x}$ more parameters to it than the decoder. We argue that a strong encoder is essential for extracting robust, disentangled representations for downstream tasks. Even with a high compression rate, a smaller decoder can still generate high-quality audio while offering faster inference, which is beneficial for streaming applications. For the decoder, we choose the more efficient Vocos [62] architecture over HiFi-GAN [32]. Vocos maintains consistent feature resolution and uses inverse STFT for upsampling, minimizing aliasing and improving computational efficiency. The decoder processes features through ConvNeXt [38] blocks and projects the sequence of hidden representations to Fourier coefficients for waveform reconstruction. The final audio is synthesized using inverse STFT.

**Discriminator.** Following HiFi-GAN [32], we employ a multi-period discriminator and a multi-scale discriminator. This approach slightly differs from prior work [83, 14, 62, 34, 25], which utilize multi-resolution and/or STFT-based discriminators in place of a multi-scale discriminator. The multi-resolution and STFT-based discriminators are particularly useful for mitigating over-smoothing artifacts in high-frequency components [34], which are more critical for music and environmental sounds. Since our focus is on speech (i.e. medium frequency range), we stick to the simpler HiFi-GAN setup.

## 3.2 Training

The training process consists of two stages. In the **first stage**, the compressor, quantizer, and decompressor are jointly trained to reconstruct the encoder continuous representations, ensuring that the tokens retain both semantic and acoustic information from the encoder, which is kept frozen. The training objective includes reconstruction loss and entropy loss. The reconstruction loss is computed as the squared L2 distance between the reconstructed and original encoder features. The entropy loss, defined as in [82, 87], encourages both confident predictions and uniform code utilization. Note that we omit the commitment loss used in standard VQ, as for BSQ there is no concern of embedding divergence (quantization error is bounded).

In the **second stage**, the decoder is trained to resynthesize audio from the encoder *continuous* representations. The training objective includes adversarial loss, reconstruction loss, and feature matching loss, as in [32]. However, following [83], we use a hinge loss formulation instead of least squares. The reconstruction loss is computed as the L1 distance between the reconstructed and original log-Mel spectrograms, while the feature matching loss is the mean of the distances between the $l$-th feature maps of the $k$-th subdiscriminator.

This design allows the second stage to run in parallel with the first, simplifying the training setup. At inference, the same decoder operates on dequantized features produced by the compressor-quantizer-decompressor pipeline. Because the decompressor is trained to reconstruct the original continuous features from the discrete codes, these dequantized features closely approximate the originals. As a result, the decoder maintains strong performance even when using dequantized features as input, without requiring any additional fine-tuning.

This decoupled training approach ensures that both semantic and acoustic information are preserved in the tokens, which is crucial for downstream tasks while maintaining high reconstruction quality. If trained end-to-end without additional constraints on the hidden representations (e.g. distillation loss), the reconstruction loss prioritizes acoustic features, as observed in [14, 34].

## 4 Experiments

### 4.1 FocalCodec

We train FocalCodec on LibriTTS [84], resampled to 16 kHz. We train three variants of the model with a codebook size of 8192 and token rates of 50 Hz, 25 Hz, and 12.5 Hz by adjusting the temporal downsampling factors in the compressor layers to (1, 1, 1), (2, 1, 1), and (2, 2, 1), respectively. These patterns are mirrored in the decompressor layers for upsampling. Information about hyperparameters and training details can be found in Section E.1.

## 4.2 Baselines

We compare our models to recent state-of-the-art low-bitrate codecs across acoustic, semantic, and hybrid categories. Since this paper focuses on low-bitrate codecs, when multiple quantizers are available, we configure them to achieve a bitrate below 1.50 kbps, ensuring a fair comparison. For acoustic codecs, we compare against **EnCodec** [14], **DAC**[3] [34], **WavTokenizer** [25], and **BigCodec** [76]. Among these, BigCodec is the current state-of-the-art for low-bitrate

Table 1: Codecs considered in our experimental analysis.

| Codec | Bitrate (kbps) | Sample Rate (kHz) | Token Rate (Hz) | Codebooks | Code Size | Params (M) | MACs (G) |
|---|---|---|---|---|---|---|---|
| EnCodec | 1.50 | 24 | 75.0 | $2 \times 1024$ | 128 | 15 | 2 |
| DAC | 1.00 | 16 | 50.0 | $2 \times 1024$ | 8 | 74 | 56 |
| WavLM6-KM | 0.45 | 16 | 50.0 | $1 \times 512$ | 1024 | 127 | 28 |
| SpeechTokenizer | 1.00 | 16 | 50.0 | $2 \times 1024$ | 1024 | 108 | 17 |
| SemantiCodec | 0.65 | 16 | 25.0 | $2 \times 8192$ | 1536 | 1033 | 1599 |
| Mimi | 0.69 | 24 | 12.5 | $5 \times 2048$ | 256 | 82 | 11 |
| WavTokenizer | 0.48 | 24 | 40.0 | $1 \times 4096$ | 512 | 85 | 3 |
| BigCodec | 1.04 | 16 | 80.0 | $1 \times 8192$ | 8 | 160 | 61 |
| Stable Codec | 0.70 | 16 | 25.0 | $2 \times 15625$ | 6 | 950 | 37 |
| **FocalCodec @50** | 0.65 | 16 | 50.0 | $1 \times 8192$ | 13 | 142 | 9 |
| **FocalCodec @25** | 0.33 | 16 | 25.0 | $1 \times 8192$ | 13 | 144 | 9 |
| **FocalCodec @12.5** | 0.16 | 16 | 12.5 | $1 \times 8192$ | 13 | 145 | 8 |

speech reconstruction quality [74]. We use the official checkpoints for these models. We do not include the recent TS3-Codec [74], which matches BigCodec performance at an even lower bitrate, as it is not publicly available. However, we contacted the authors to request reconstructed samples for comparison. Additional results related to TS3-Codec can be found in Section G.1.

For semantic codecs, we adopt the approach introduced in [71], which quantizes layer-6 representations from WavLM-large using k-means clustering with 512 centroids. These representations are fed into a Conformer [19] encoder to reconstruct continuous representations, followed by a HiFi-GAN decoder. This baseline, referred to as **WavLM6-KM**, provides a direct comparison between our codec and another model leveraging WavLM layer-6 features but differing in design and training methodology. Since the code and checkpoints for WavLM6-KM are not publicly available, we reimplemented the model using a subset of LibriSpeech [47]. Note that we do not include additional baselines from this category, as semantic codecs typically underperform in terms of reconstruction quality [48] or require much higher bitrates to be competitive in this regard [43]. Furthermore, most hybrid codecs are already built on top of semantic representations. Therefore, we prioritize the hybrid category, to which our codec also belongs. For hybrid codecs, we compare against **SpeechTokenizer** [86], **SemantiCodec** [37], **Mimi** [16], and **Stable Codec** [48], using their official checkpoints. The configurations and details of each model are summarized in Table 1. Multiply-accumulate operations per second (MACs) are measured using `ptflops`[4]. Additional information about the baselines is provided in Section D.

## 4.3 Speech Resynthesis

We evaluate FocalCodec on speech resynthesis, considering both English and multilingual speech. For English speech, we use **LibriSpeech** [47] `test-clean`. For multilingual speech, following [76], we randomly select 100 utterances from each of the 7 foreign languages in **Multilingual LibriSpeech** [51] (Dutch, French, German, Italian, Polish, Portuguese, and Spanish), resulting in a total of 700 utterances[5]. We also consider the more realistic scenario of speech contaminated with environmental noise. For this, we use the test splits of **VoiceBank** [66] and the more challenging **Libri1Mix**, which is constructed by mixing clean utterances from the first speaker of LibriMix [12] with noise from WHAM! [73].

We evaluate the models using objective metrics. To measure naturalness, we employ **UTMOS** [59] for clean speech and **DNSMOS** [56] for noisy speech. Note that we do not include signal-level metrics such as SNR, PESQ [58], or STOI [64], as these metrics do not correlate well with perceived reconstruction quality [48, 71]. To evaluate speaker fidelity, we compute the cosine similarity (**Sim**) between speaker embeddings extracted from the reconstructed audio and the target audio. These embeddings are obtained using WavLM-base-SV[6] [9]. To assess intelligibility, we compute the differential word error rate (**dWER**) [72], which measures the difference in word error rate between

---

[3]Note that we use the 16 kHz checkpoint, whereas the original results in [34] use the 24 kHz checkpoint.

[4]https://pypi.org/project/ptflops/0.7.4/

[5]https://zenodo.org/records/14791114

[6]https://huggingface.co/microsoft/wavlm-base-sv

Table 2: Speech resynthesis performance across clean and noisy datasets.

| Codec | Bitrate (kbps)↓ | UTMOS↑ | dWER↓ | Sim↑ | Code Usage↑ | Norm. Entropy↑ | RTF↑ | DNSMOS↑ | dWER↓ | Sim↑ | Code Usage↑ | Norm. Entropy↑ | RTF↑ |
|---|---|---|---|---|---|---|---|---|---|---|---|---|---|
| | | *Clean – LibriSpeech test-clean* | | | | | | *Noisy – VoiceBank* | | | | | |
| Reference | — | 4.09 | 0.00 | 100.0 | — | — | — | 3.56 | 0.00 | 100.0 | — | — | — |
| EnCodec | 1.50 | 1.58 | 8.08 | 93.8 | 93.4 | 82.1 | 109 | 2.76 | 28.16 | 87.7 | 77.5 | 78.1 | 44 |
| DAC | 1.00 | 1.29 | 20.04 | 89.2 | **100.0** | 91.7 | 89 | 2.72 | 63.90 | 79.8 | 98.7 | 88.4 | 48 |
| WavLM6-KM | 0.45 | 3.75 | 6.20 | 90.0 | 26.4 | 95.4 | 85 | 3.06 | 20.67 | 82.9 | 24.8 | 92.3 | 44 |
| SpeechTokenizer | 1.00 | 2.28 | 5.14 | 91.6 | 95.9 | 97.0 | 63 | 2.74 | 34.51 | 82.2 | 88.1 | 88.4 | 42 |
| SemantiCodec | 0.65 | 2.91 | 8.97 | 96.0 | 75.9 | 94.4 | 0.62 | 3.13 | 31.46 | 90.6 | 52.4 | 92.6 | 0.28 |
| Mimi | 0.69 | 3.29 | 5.73 | 96.0 | 95.6 | 91.8 | 137 | 3.01 | 28.00 | 87.8 | 78.6 | 85.5 | 47 |
| WavTokenizer | 0.48 | 3.78 | 11.55 | 95.4 | **100.0** | 96.7 | 181 | 3.09 | 42.12 | 89.8 | 94.8 | 94.0 | 63 |
| BigCodec | 1.04 | 4.11 | 2.55 | **98.5** | **100.0** | 98.6 | 22 | 3.19 | 20.67 | **92.3** | 99.8 | **96.8** | 17 |
| Stable Codec | 0.70 | **4.32** | 4.97 | 94.7 | 98.5 | 94.7 | 103 | **3.33** | 20.32 | 88.8 | 75.7 | 95.4 | 39 |
| **FocalCodec@50** | 0.65 | 4.05 | **2.18** | 97.4 | **100.0** | **98.9** | 185 | 3.16 | **8.08** | 91.3 | 98.0 | 96.2 | 80 |
| **FocalCodec@25** | 0.33 | 4.14 | 3.30 | 96.3 | 99.8 | 98.4 | 195 | 3.17 | 11.75 | 90.1 | 89.6 | 96.0 | **81** |
| **FocalCodec@12.5** | **0.16** | 4.22 | 7.94 | 93.9 | 98.2 | 97.4 | **208** | 3.22 | 27.97 | 84.7 | 77.3 | 95.5 | 79 |
| | | *Clean – Multilingual LibriSpeech 700* | | | | | | *Noisy – Libri1Mix* | | | | | |
| Reference | — | 2.84 | 0.00 | 100.0 | — | — | — | 3.73 | 0.00 | 100.0 | — | — | — |
| EnCodec | 1.50 | 1.33 | 29.60 | 95.5 | 93.4 | 79.2 | 140 | 2.40 | 55.17 | 86.3 | 84.4 | 78.7 | 97 |
| DAC | 1.00 | 1.24 | 56.08 | 89.1 | **100.0** | 90.0 | 97 | 2.40 | 90.92 | 76.6 | 99.1 | 88.8 | 91 |
| WavLM6-KM | 0.45 | 2.97 | 44.54 | 89.5 | 28.1 | 0.91 | 125 | 2.87 | 36.60 | 85.9 | 26.8 | 95.5 | 65 |
| SpeechTokenizer | 1.00 | 1.55 | 56.32 | 92.0 | 96.1 | 94.0 | 74 | 2.58 | 57.26 | 82.8 | 93.5 | 96.5 | 63 |
| SemantiCodec | 0.65 | 1.87 | 36.21 | 97.7 | 76.4 | 94.7 | 0.74 | 2.67 | 51.18 | 89.9 | 64.7 | 90.8 | 91 |
| Mimi | 0.69 | 2.08 | 30.96 | 96.7 | 95.9 | 89.0 | 239 | 2.65 | 49.14 | 89.4 | 90.8 | 90.1 | 104 |
| WavTokenizer | 0.48 | 2.64 | 49.73 | 97.0 | 97.6 | 95.4 | 290 | 2.53 | 70.10 | 86.3 | 96.4 | 95.4 | **165** |
| BigCodec | 1.04 | 2.86 | 15.24 | **99.1** | **100.0** | 97.9 | 24 | 2.75 | 53.26 | 88.3 | **100.0** | 98.2 | 19 |
| Stable Codec | 0.70 | **3.47** | 56.99 | 95.9 | 92.9 | 93.8 | 144 | 2.91 | 43.52 | 90.0 | 95.8 | 93.4 | 68 |
| **FocalCodec@50** | 0.65 | 2.96 | **12.57** | 98.3 | **100.0** | 98.1 | 269 | **2.93** | **27.89** | 91.6 | **100.0** | **98.5** | 155 |
| **FocalCodec@25** | 0.33 | 3.16 | 19.78 | 97.3 | 99.2 | 97.4 | 292 | 2.91 | 34.27 | 90.7 | 99.6 | 97.9 | 161 |
| **FocalCodec@12.5** | **0.16** | 3.37 | 54.15 | 95.2 | 96.4 | 96.9 | **296** | 2.92 | 42.59 | 88.9 | 97.2 | 97.2 | **164** |

the reconstructed and target audio, using transcriptions from Whisper small[7] [53]. To ensure fairness in evaluation, we do not use more powerful ASR models (e.g. Whisper large-v3), as these models can correct pronunciation mistakes and are more robust to noise, potentially hiding flaws in the reconstruction. We also report **code usage**, i.e. the ratio of unique tokens used to the codebook size (averaged over codebooks for multi-codebook models), and **normalized entropy** [13, 48], where higher values indicate more uniform codebook usage. For inference speed, we measure the real-time factor (**RTF**), i.e. the ratio of the reconstructed audio duration to the processing time. An RTF greater than 1 indicates faster-than-real-time performance, measured on an NVIDIA V100 GPU with 32 GB of memory.

Results are presented in Table 2. FocalCodec shows strong performance across both clean and noisy speech resynthesis tasks. On clean speech, FocalCodec@50 achieves the best trade-off of quality, intelligibility, and efficiency. Notably, FocalCodec is the best in terms of dWER, surpassing BigCodec, which is currently state-of-the-art. It also generalizes well to multilingual speech, obtaining the lowest dWER and high Sim. Note that FocalCodec, WavLM6-KM, SpeechTokenizer, BigCodec and Stable Codec were trained exclusively on English speech. In noisy speech resynthesis, FocalCodec@50 again excels, achieving the lowest dWER by a large margin on both VoiceBank and Libri1Mix, while maintaining high speaker similarity. Meanwhile, FocalCodec@25 and FocalCodec@12.5 exhibit some degradation in dWER and speaker similarity, particularly in multilingual settings, due to their significantly lower bitrates. Nevertheless, despite operating at just 0.16 kbps, FocalCodec@12.5 remains competitive with several baselines that use much higher bitrates (e.g. EnCodec). It is also worth noting that FocalCodec's UTMOS tends to increase at lower bitrates, likely due to the stronger smoothing effect introduced by downsampling. However, UTMOS tends to saturate and may not fully capture perceptual quality [59]. dWER and Sim are therefore essential to provide a more comprehensive evaluation. Finally, the high code usage and normalized entropy across all FocalCodec variants indicate efficient token utilization, contributing to their strong overall performance. Additional results on reconstruction quality, including subjective evaluations, streamability and Mel-spectrogram analysis, can be found in Sections G.2 to G.4.

## 4.4 Voice Conversion

We conduct one-shot voice conversion experiments to verify that FocalCodec can effectively disentangle speaker information from content despite its single-codebook design. This task involves

---

[7]https://huggingface.co/openai/whisper-small

converting speech from a source speaker to an arbitrary target speaker using reference speech from the target speaker. For single-codebook baselines, including FocalCodec, we use $k$-nearest neighbors search in the codec feature space, as in [2]. Specifically, we replace each frame in the reconstructed feature sequence (right before the decoder) with the average of the $k = 4$ closest matches in terms of cosine distance from continuous features extracted from the reference. For multi-codebook baselines, instead, we follow the procedure in [86]. The source and reference speech are to-

Table 3: One-shot voice conversion on VCTK [77].

| Codec | Bitrate (kbps) ↓ | UTMOS ↑ | dWER ↓ | Sim ↑ | RTF ↑ |
|---|---|---|---|---|---|
| Reference | — | 4.09 | 0.00 | 100.0 | — |
| EnCodec | 1.50 | 1.24 | 86.52 | 72.2 | 57 |
| DAC | 1.00 | 1.25 | 104.00 | 67.2 | 60 |
| WavLM6-KM | 0.45 | 2.90 | 26.68 | 92.4 | 57 |
| SpeechTokenizer | 1.00 | 1.49 | **20.32** | 81.2 | 33 |
| SemantiCodec | 0.65 | 2.02 | 106.00 | 72.8 | 0.60 |
| Mimi | 0.69 | 2.40 | 110.00 | 89.7 | 71 |
| WavTokenizer | 0.48 | 3.13 | 43.15 | 73.4 | 89 |
| BigCodec | 1.04 | 1.31 | 99.96 | 68.9 | 13 |
| Stable Codec | 0.70 | **3.76** | 27.63 | 71.1 | 65 |
| **FocalCodec@50** | 0.65 | 3.38 | 21.27 | 92.2 | 116 |
| **FocalCodec@25** | 0.33 | 3.40 | 23.59 | **92.6** | **118** |
| **FocalCodec@12.5** | **0.16** | 3.43 | 29.93 | **92.6** | 117 |

kenized, and the first codebook tokens from the source are concatenated with the second-to-last codebook tokens from the reference. The resulting sequence is then forwarded to the decoder. If sequence lengths differ, the reference is truncated or circularly padded as needed. Effective disentanglement of content and speaker information between first and subsequent codebooks is expected to yield fair voice conversion performance. We conduct voice conversion experiments on **VCTK** [77], which includes parallel utterances from different speakers. To create the test set, we randomly select an utterance from a source speaker, the corresponding utterance from a target speaker, and an utterance with different content from the same target speaker to act as the reference. Among available reference utterances, we select the longest to minimize padding issues. We repeat this process for each speaker, for each of the ∼24 parallel utterances, resulting in a dataset with 2521 samples. To evaluate performance, we use UTMOS, dWER, Sim, and RTF as defined in Section 4.3.

As reported in Table 3, FocalCodec achieves the highest speaker similarity while maintaining good intelligibility, confirming its suitability for voice conversion tasks. This is particularly impressive, especially compared to other hybrid codecs like SpeechTokenizer and Mimi, which are explicitly optimized to disentangle semantic information in the first codebook and acoustic information in the following. Despite this, FocalCodec outperforms these models, excelling in both speaker identity preservation and intelligibility, striking a remarkable balance of quality, efficiency, and speaker similarity. WavLM6-KM ranks as the second-best performing model, which is expected since it shares the same encoder as FocalCodec. In contrast, acoustic codecs struggle with this task, as they do not separate speaker and content information.

## 4.5 Downstream Tasks

To evaluate the quality of the learned discrete representations, we train downstream models on both discriminative and generative tasks.

**Discriminative Tasks.** We evaluate performance on automatic speech recognition (ASR), speaker identification (SI), and speech emotion recognition (SER). These tasks allow us to assess token quality along three axes: semantic information retention (ASR), acoustic information retention (SI), and emotion information retention (SER, which requires a non-trivial combination of semantic and acoustic clues). To focus on the disentanglement of learned representations, we employ shallow downstream models, aiming to stay as close as possible to linear probing. Following [86], we employ a shallow BiLSTM for all tasks. For **ASR**, we use LibriSpeech [47] `train-clean-100` and `train-clean-360` for training, `dev-clean` for validation, and `test-clean` for testing. The word error rate (WER) is reported. For **SI**, we also use LibriSpeech, grouping utterances from `train-clean-100` and `train-clean-360` by speaker ID. Data are randomly split into training, validation and test sets in a ratio of 80% / 10% / 10%. The speaker error rate (ER) is reported. For **SER**, we use the IEMOCAP dataset [7], focusing on four emotions: sadness, happiness, anger, and neutral. Sessions 1-4 are used for training, session 5F for validation, and session 5M for testing. The emotion ER is reported. Details about the model architecture, hyperparameters, and training procedure are provided in Section E.2.

Table 4 shows the results. In ASR, FocalCodec@50 achieves the third lowest WER. While SpeechTokenizer and Stable Codec perform slightly better, the former operates at ∼1.5x higher bitrate using

Table 4: Evaluation on discriminative and generative downstream tasks.

| Codec | Bitrate (kbps) ↓ | Discriminative Tasks | | | Generative Tasks | | | | | | | | |
|---|---|---|---|---|---|---|---|---|---|---|---|---|---|
| | | ASR | SI | SER | SE | | | SS | | | TTS | | |
| | | WER ↓ | ER ↓ | ER ↓ | DNSMOS ↑ | dWER ↓ | Sim ↑ | DNSMOS ↑ | dWER ↓ | Sim ↑ | UTMOS ↑ | dWER ↓ | Sim ↑ |
| Reference | — | — | — | — | 3.56 | 0.00 | 100.0 | 3.77 | 0.00 | 100.0 | 4.09 | 0.00 | 100.0 |
| EnCodec | 1.50 | 27.89 | 3.00 | 47.00 | 3.11 | 37.10 | 85.9 | 3.11 | 78.51 | 87.3 | 1.71 | 64.28 | 83.2 |
| DAC | 1.00 | 35.89 | 3.27 | 45.90 | 3.03 | 67.65 | 81.7 | 2.76 | 106.00 | 83.3 | 1.34 | 47.06 | 85.9 |
| WavLM6-KM | 0.45 | 19.04 | 22.30 | 42.90 | 3.52 | 22.85 | 83.6 | 3.49 | 76.91 | 85.0 | 3.74 | 38.67 | 88.7 |
| SpeechTokenizer | 1.00 | **14.97** | 2.73 | **41.50** | 3.21 | 29.82 | 85.9 | 3.13 | 83.99 | 87.3 | 2.69 | 35.46 | 89.2 |
| SemantiCodec | 0.65 | 41.42 | 15.90 | 51.60 | **3.59** | 102.00 | 83.3 | 3.59 | 123.00 | 84.4 | 2.82 | 48.38 | 91.4 |
| Mimi | 0.69 | 22.98 | 5.43 | 44.70 | 3.30 | 53.98 | 84.6 | 3.41 | 93.23 | 88.1 | 3.11 | 28.63 | **93.6** |
| WavTokenizer | 0.48 | 35.62 | 2.44 | 49.80 | 3.41 | 51.75 | 88.6 | 3.54 | 105.00 | 86.4 | 3.68 | 47.56 | 92.8 |
| BigCodec | 1.04 | 26.41 | **2.34** | 47.50 | 3.52 | 26.68 | **93.2** | 3.54 | 89.24 | **89.4** | 3.43 | 54.43 | 89.4 |
| Stable Codec | 0.70 | 16.85 | 16.50 | 46.54 | 3.55 | 35.57 | 82.8 | 3.61 | 103.00 | 78.2 | 3.19 | 49.28 | 88.8 |
| **FocalCodec@50** | 0.65 | 17.63 | 4.48 | 45.60 | 3.47 | **10.93** | 91.4 | **3.71** | 73.87 | 89.0 | 4.11 | 28.10 | 93.3 |
| **FocalCodec@25** | 0.33 | 21.12 | 6.07 | 46.80 | 3.49 | 14.74 | 90.0 | 3.69 | 99.96 | 85.4 | **4.16** | **16.75** | 91.6 |
| **FocalCodec@12.5** | **0.16** | 33.24 | 11.69 | 46.30 | 3.58 | 36.98 | 86.9 | 3.57 | 116.00 | 80.8 | 4.12 | 21.59 | 90.8 |

two codebooks, while the latter was fine-tuned on force-aligned phoneme data to enhance semantic representations. In contrast, our model is purely self-supervised. In SI, FocalCodec@50 achieves a marginally higher error rate (∼2%) than codecs such as BigCodec and WavTokenizer. However, these models perform significantly worse in ASR due to being trained solely with reconstruction-based objectives. On the other hand, the purely semantic WavLM-KM6 codec performs competitively in ASR but exhibits the highest ER in SI despite using the same encoder as FocalCodec. This further confirms the effectiveness of our codec design, as it improves WER over WavLM-KM6 while preserving speaker information. Interestingly, Stable Codec also performs poorly in SI, likely because semantic fine-tuning tends to remove acoustic information from the representations. In SER, no codec clearly excels, with FocalCodec@50 performing on par with the best models. Overall, Focal-Codec@50 shows competitive performance across all discriminative tasks, rivaling hybrid codecs with more complex multi-codebook designs and higher bitrates. The more compressed variants, FocalCodec@25 and FocalCodec@12.5, still achieve good performance while operating at ultra-low bitrates.

**Generative Tasks.** We evaluate performance on speech enhancement (SE), speech separation (SS), and text-to-speech (TTS). For these tasks, we employ more powerful transformer-based downstream models, focusing on generation quality. For **SE** we use VoiceBank [66]. To form a validation set, we randomly select two speakers from the training set. The input tokens are extracted from noisy utterances, while the target tokens come from clean utterances. Performance metrics include DNSMOS, dWER, and Sim. For **SS**, we use Libri2Mix [12] `train-100`, `dev`, and `test` sets. The setup mirrors that of speech enhancement: input tokens are derived from speech mixtures, while target tokens correspond to the two individual sources. For **TTS**, we use LibriSpeech `train-clean-100` and `train-clean-360` for training, `dev-clean` for validation, and `test-clean` for testing. Note that `test-clean` contains several utterances longer than 20 seconds (∼4%), whereas our training splits include almost none. To reduce the mismatch between training and testing conditions, we removed these long utterances from the test set. The input consists of character-based text tokens, while the target tokens are derived from the corresponding utterances. Performance is evaluated using UTMOS, dWER, and Sim. Details about the model architecture, hyperparameters, and training procedure are provided in Section E.2.

From Table 4, we observe that in SE, FocalCodec@50 significantly outperforms all other baselines in terms of dWER. A similar trend is observed for SS, where FocalCodec@50 is consistently superior to the other baselines. However, the absolute performance is still far from practical utility, likely due to the loss of information crucial for SS during quantization. As with discriminative tasks, FocalCodec@25 and FocalCodec@12.5 show degraded performance, due to their ultra-low bitrates. However, this trend is reversed for TTS, with FocalCodec@25 achieving the best overall results, followed closely by FocalCodec@12.5. This can be attributed to the fact that, in autoregressive modeling, shorter sequences reduce the computational burden and simplify the task of predicting the next token. Both models, operating at a frame rate closer to that of text with a single codebook, make next-token prediction easier and more computationally efficient than other methods. This highlights the importance of having compact representations for downstream tasks. Note, however, that we trained on only 460 hours of speech, which explains why TTS performance is not state-of-the-art.

### 4.6 Ablation Studies

Due to limited computational resources, we perform ablation studies on a smaller variant of Focal-Codec. This variant is similar to the 50 Hz model, with the main difference being the model size, as detailed in Section E.1. We focus on the clean speech resynthesis task using LibriSpeech `test-clean` [47]. The results are shown in Table 5.

Table 5: Ablation studies on LibriSpeech `test-clean` [47].

| Compression Block | Downscale Activation | Quantizer | UTMOS ↑ | dWER ↓ | Sim ↑ |
|---|---|---|---|---|---|
| Focal modulation | Snake | BSQ | 3.73 | **2.54** | **95.7** |
| Focal modulation | Snake | **FSQ** | 3.71 | 2.61 | 94.8 |
| Focal modulation | Snake | **LFQ** | **3.74** | 2.75 | 95.4 |
| Focal modulation | **Leaky ReLU** | LFQ | 3.72 | 2.85 | 95.2 |
| **Conformer** | **Snake** | LFQ | **3.74** | 3.58 | 94.3 |
| **AMP** | Snake | LFQ | 3.70 | 4.52 | 94.3 |
| **Linear** | Snake | LFQ | 2.55 | 9.37 | 82.5 |

Replacing BSQ with FSQ [41] leads to worse UTMOS, dWER, and Sim. It also results in less uniform code usage, as evidenced by the normalized entropy measured for these two configurations (99.7 for BSQ vs. 97.7 for FSQ). Replacing BSQ with vanilla LFQ results in worse dWER and Sim despite similar UTMOS. Replacing Snake activations with leaky ReLU causes only minor performance degradation. The most significant performance drop occurs when the focal modulation blocks are replaced with Conformer [19] blocks, anti-aliased multi-periodicity (AMP) [35] blocks, or linear layers, in this order. This leads to a notable decrease in both dWER and Sim. This analysis further validates our design choices, highlighting the importance of the selected components for achieving optimal performance.

## 5 Conclusions

In this work, we introduced FocalCodec, a low-bitrate single-codebook speech codec that employs a novel architecture based on focal modulation. It delivers competitive performance in speech resynthesis and voice conversion at low and ultra-low bitrates while maintaining robustness across diverse conditions, including multilingual and noisy speech. Furthermore, it effectively preserves both semantic and acoustic information, providing powerful discrete representations for downstream tasks. A detailed discussion of the limitations and societal impact of this work is provided in Sections A and B.

## 6 Acknowledgments

We acknowledge support from NSERC, the Digital Research Alliance of Canada (alliancecan.ca), the NVIDIA Academic Grant Program for computing resources, and Translated for funding through the Immediate research grant.

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

# A  Limitations

Despite its competitive performance, FocalCodec is undertrained compared to other state-of-the-art approaches. While the WavLM encoder benefits from 94k hours of pretraining, the rest of the pipeline was trained on only a few hundred hours of clean English speech. Expanding the dataset to include more data, a broader range of domains (e.g. multilingual speech, mixtures, etc.) could further improve quality, robustness, and versatility of the model. By comparison, competing methods such as WavTokenizer (8k hours), StableCodec (105k hours), and Mimi (7M hours) are trained on significantly larger and more diverse datasets.

# B  Societal Impact

We believe this research has the potential for meaningful societal benefits. Ultra-low bitrate speech codecs can significantly reduce the bandwidth and storage requirements for transmitting and storing spoken content. This has practical implications for improving the accessibility and efficiency of voice communication in bandwidth-constrained settings, such as rural or remote areas, and for enabling on-device speech applications with minimal resource consumption. However, we also acknowledge potential risks associated with misuse. In particular, voice conversion capabilities enabled by FocalCodec could potentially be exploited for malicious purposes, including voice cloning, impersonation, and the creation of deceptive or harmful deepfake audio content. To mitigate these risks, we encourage responsible use of this technology and further research into detection and authentication mechanisms to ensure secure and ethical deployment. It is worth noting nevertheless that similar capabilities are already publicly accessible, with models like https://huggingface.co/amphion/Vevo offering few-shot voice conversion.

# C  Datasets

The following datasets were used in this work:

- **LibriSpeech** [47] is a large-scale corpus of English read speech derived from audiobooks in the LibriVox project. It contains approximately 1000 hours of speech sampled at 16 kHz, with predefined training, validation, and test splits. License: CC BY 4.0.

- **LibriTTS** [84] is a corpus designed for text-to-speech research, constructed from the same source as LibriSpeech. It consists of 585 hours of transcribed speech with predefined training, validation, and test splits. License: CC BY 4.0.

- **Multilingual LibriSpeech** [51] is an extension of LibriSpeech to multiple languages, including English, German, Dutch, French, Spanish, Italian, Portuguese and Polish. It provides approximately 44,500 hours of transcribed English speech and about 6000 hours from other languages. License: CC BY 4.0.

- **VoiceBank** [66] is a dataset primarily used for speech enhancement, including 11,572 utterances from 28 speakers in the training set (noise at 0 dB, 5 dB, 10 dB, and 15 dB), and 872 utterances from 2 unseen speakers in the test set (noise at 2.5 dB, 7.5 dB, 12.5 dB, and 17.5 dB). License: CC BY 4.0.

- **LibriMix** [12] is a dataset for speech separation and enhancement, created by mixing LibriSpeech utterances with noise from the WHAM! [73] corpus. It provides mixtures of two or three speakers at different signal-to-noise ratios. License: MIT.

- **VCTK** [77] is a corpus of English speech recordings from 110 speakers with various accents. It is widely used for speaker adaptation, text-to-speech, and voice conversion tasks. License: CC BY 4.0.

- **IEMOCAP** [7] is a dataset designed for emotion recognition, consisting of scripted and improvised dialogues performed by 10 actors. It includes audio, video, and textual transcriptions with emotion labels such as happiness, sadness, and anger. License: https://sail.usc.edu/iemocap/iemocap_release.htm.

# D  Baselines

Additional information about the baseline codecs is provided in Table 6. For our WavLM6-KM [71] reproduction, we use LibriSpeech `train-clean-100` and `train-clean-360`. First, we train a k-means quantizer with 512 centroids on top of layer-6 representations from WavLM-large. We train on audio chunks of 16,000 samples with a large batch size of 512 for improved stability, and we stop training when cluster centroids stop changing significantly. Then, we train a dequantizer to minimize the L2 loss between quantized and original WavLM features. We employ a Conformer [19] encoder with 6 layers, 4 attention heads, a hidden dimension of 512, and a feed-forward layer dimension of 512. We train on audio chunks of 7040 samples with a batch size of 16. We use the AdamW [40] optimizer with an initial learning rate of 0.0005, $\beta_1$ of 0.8, $\beta_2$ of 0.99, weight decay of 0.01, and dropout of 0.1. The learning rate is reduced by a factor of 0.9 if validation loss does not improve within a margin of 0.0025. Gradients are clipped to a maximum L2 norm of 5. Training stops when validation

Table 6: Baseline codecs.

| Codec | Causal | Training Datasets | Hours | Multilingual | Audio Domain | Checkpoint | License |
|---|---|---|---|---|---|---|---|
| EnCodec [14] | Optional | DNS, CommonVoice, AudioSet, FSD50K, Jamendo | 17k+ | Yes | General | encodec_24khz | MIT |
| DAC [34] | No | DAPS, DNS, CommonVoice, VCTK, MUSDB, Jamendo | 10k+ | Yes | General | weights_16khz.pth | MIT |
| WavLM6-KM [71] | No | Subset of LibriSpeech (in addition to Libri-Light, GigaSpeech, and VoxPopuli English for WavLM pretraining) | 460 (+ 94k) | No | Speech | discrete-wavlm-codec | Apache 2.0 |
| SpeechTokenizer [86] | No | LibriSpeech | 960 | No | Speech | speechtokenizer_hubert_avg | Apache 2.0 |
| SemantiCodec [37] | No | GigaSpeech, subset of OpenSLR, Million Song Dataset, MedleyDB, MUSDB18, AudioSet, WavCaps, VGGSound | 20k+ | Yes | General | semanticodec_tokenrate_50 | MIT |
| Mimi [16] | Yes | Predominantly English speech (in addition to Libri-Light, GigaSpeech, and VoxPopuli English for WavLM pretraining) | 7M (+ 94k) | Likely | Speech | mimi | CC BY 4.0 |
| WavTokenizer [25] | No | LibriTTS, VCTK, subset of CommonVoice, subset of AudioSet, Jamendo, MUSDB | 8k | Yes | General | WavTokenizer-large-unify-40token | MIT |
| BigCodec [76] | No | LibriSpeech | 960 | No | Speech | bigcodec.pt | MIT |
| Stable Codec [48] | Optional | Libri-Light, Multilingual LibriSpeech English | 105k | No | Speech | stable-codec-speech-16k | StabilityAI |

loss does not decrease for several consecutive epochs. Finally, we train a HiFi-GAN V1 [32] decoder on audio chunks of 7040 samples with a batch size of 16. We use the AdamW optimizer with an initial learning rate of 0.0002, $\beta_1$ of 0.8, $\beta_2$ of 0.99, and weight decay of 0.01. The learning rate follows an exponential decay schedule with a factor of 0.999. Training continues until perceived audio quality stops improving.

# E  Hyperparameters and Training Details

## E.1  FocalCodec

The compressor processes 1024-dimensional WavLM features and forwards them through 3 focal downscaling blocks with hidden dimensions of 1024, 512, and 256, respectively. Each block has two focal levels, a window size of 7, a focal factor of 2, and a layer scale initialization of 0.0001. A final projection maps the 256-dimensional hidden states to latent representations of dimension 13, which are then quantized with a binary spherical codebook of $2^{13} = 8192$ codes. The decompressor mirrors the compressor, replacing focal downscaling blocks with focal upscaling blocks to reconstruct the 1024-dimensional continuous representations from the quantized latent codes. We use a weight of 1.0 for the reconstruction loss and a weight of 0.1 for the entropy loss. We train on LibriTTS [84] (585 hours from 2456 speakers) using full utterances rather than fixed-length chunks, which differs from related work. This approach allows us to fully exploit the unlimited receptive field of focal modulation. This is in line with our vision that the encoder should be as powerful as possible to extract high-quality representations, while the decoder can be lightweight and use limited context windows. For this stage, we use the AdamW [40] optimizer with an initial learning rate of 0.0005, $\beta_1$ of 0.8, $\beta_2$ of 0.99, and weight decay of 0.01. The learning rate is reduced by a factor of 0.9 if validation loss does not improve within a margin of 0.0025. Gradients are clipped to a maximum L2 norm of 5. Training stops when validation loss does not decrease for several consecutive epochs.

The decoder processes 1024-dimensional WavLM features and forwards them through 8 ConvNeXt blocks with a hidden dimension of 512, a feed-forward dimension of 1536, a kernel size of 7, and padding of 3. For the STFT, we set the FFT size to 1024 samples and the hop length to 320. The feature matching loss is calculated using 80-dimensional log-Mel spectrograms with the same STFT configuration. The discriminator adopts the convolutional architecture introduced in [32]. We train on LibriTTS using audio chunks of 7040 samples with a batch size of 16. Due to resource constraints, our training is limited to the `train-clean-100` split. We found this amount of data sufficient to obtain high-quality reconstructions. We use the AdamW optimizer with an initial learning rate of 0.0002, $\beta_1$ of 0.8, $\beta_2$ of 0.99, and weight decay of 0.01. The learning rate follows an exponential decay schedule with a factor of 0.999. Training continues until perceived audio quality stops improving, which occurs around 3M steps.

For the smaller variant of FocalCodec used in the ablation studies, we employ the same setup with the following modifications: the hidden sizes in the three focal downscaling blocks are reduced from 1024, 512, 256 to 512, 256, 128; the codebook size is decreased to 1024; we use HiFi-GAN-V1 [32] decoder instead of Vocos [62]; the model is trained on LibriSpeech `train-clean-100` using a batch size of 4.

## E.2  Downstream Tasks

**Automatic Speech Recognition (ASR).** The model architecture is a 2-layer BiLSTM with 512-dimensional hidden states. A CTC [18] head is stacked on top and trained to predict either characters or BPE units. Experiments use characters and BPE vocabularies of sizes 250, 500, and 1000, with the best result reported. Note that for Mimi and FocalCodec@12.5, training on characters is infeasible due to the low token rate (12.5 Hz), which results in hidden sequences shorter than the target, making them incompatible with CTC loss. For all models except Mimi, performance improves monotonically with increasing BPE sizes up to 1000,

while Mimi achieves the best results with BPE-500. If the codec employs multiple codebooks, we compute a weighted sum of the embeddings from each codebook, with the weights learned during training, as done in [9]. The embedding layer is initialized using the discrete embeddings from the codec quantizer.

**Speaker Identification (SI).**   The SI setup closely mirrors that of ASR. The only difference is that the BiLSTM output sequence is aggregated using statistics pooling, followed by a cross-entropy classification head.

**Speech Emotion Recognition (SER).**   The SER setup is the same as SI, where only the number of output classes is different.

**Speech Enhancement (SE).**   The model architecture is a Conformer [19] encoder with 6 layers, 4 attention heads, a model dimension of 512, and a feed-forward layer dimension of 2048. Codecs with multiple codebooks use a weighted sum of embeddings for the input, with independent linear heads for each codebook in the output. The embedding layer is initialized using the discrete embeddings from the codec quantizer. Training is performed using cross-entropy loss between predicted and target tokens.

**Speech Separation (SS).**   The SS setup closely mirrors that of SE. The only difference is that training is performed using cross-entropy loss with permutation invariant training [31], and the number of output heads is doubled to account for predicting two sources in parallel.

**Text-To-Speech (TTS).**   The model architecture is an autoregressive Llama 3 [17] decoder with 12 layers, 4 attention heads, 1 key-value head, a model dimension of 512, a feed-forward layer dimension of 2048, and a base RoPE frequency of 10,000. To provide speaker information, we extract speaker embeddings from the target utterance using WavLM-base [9], fine-tuned for speaker verification. The pooled speaker embedding is prepended to the text embeddings to condition the model on speaker identity. The embedding layer is initialized using the discrete embeddings from the codec quantizer. Training is performed with next-token prediction, where the input sequence consists of pooled speaker embedding, text embeddings, and speech token embeddings. The cross-entropy loss is computed only on speech tokens, while the text and speaker embeddings are excluded from loss computation. For inference, we use top-$p$ sampling with $p = 0.9$ and temperature of 1.0. Following the experimental protocol of [65], we generated 5 samples per utterance and selected the one with the lowest WER relative to the input text, using Whisper-small [53] to obtain the transcription.

**Training Details.**   For all tasks, we use AdamW [40] optimizer with a batch size of 16, an initial learning rate of 0.0001, $\beta_1 = 0.8$, $\beta_2 = 0.99$, weight decay of 0.01, and dropout of 0.1. The learning rate is reduced by a factor of 0.9 if validation loss does not improve within a margin of 0.0025. Gradients are clipped to a maximum L2 norm of 0.01. Training stops if validation loss does not decrease for several consecutive epochs.

# F   Implementation and Hardware

Software for the experimental evaluation was implemented in Python using the SpeechBrain [55, 54] toolkit. Each model is trained on a single GPU, with the choice between V100 GPUs (16 or 32 GB) and A100 GPUs (40 GB), depending on cluster resource availability.

# G   Additional Results

## G.1   Comparison to TS3-Codec

TS3-Codec [74] is a recent transformer-only architecture designed for low-bitrate streaming speech coding. Despite its lower bitrate and streamable architecture, it remains competitive with BigCodec, the current state-of-the-art. Like FocalCodec, it utilizes a single quantizer. However, its fully transformer-based architecture prioritizes reconstruction, focusing on acoustic representations. The model was trained on Libri-Light [29]. Since the model is not publicly available, we reached out to the authors to obtain reconstructions of the LibriSpeech `test-clean` for comparison. Table 7 shows the results. FocalCodec@50 surpasses TS3-Codec across all evaluated metrics, while FocalCodec@25, despite operating at a significantly lower bitrate, still achieves superior performance in terms of UTMOS and dWER. These findings further highlight the effectiveness of the proposed models.

Table 7: Clean speech resynthesis on LibriSpeech `test-clean` [47].

| Codec | Bitrate (kbps) | Sample Rate (kHz) | Token Rate (Hz) | Codebooks | Code Size | Params (M) | MACs (G) | UTMOS ↑ | dWER ↓ | Sim ↑ |
|---|---|---|---|---|---|---|---|---|---|---|
| Reference | — | — | — | — | — | — | — | 4.09 | 0.00 | 100.0 |
| TS3-Codec (X2) | 0.85 | 16 | 50.0 | 1 × 131072 | 16 | 204 | 8 | 3.84 | 4.51 | 97.1 |
| **FocalCodec@50** | 0.65 | 16 | 50.0 | 1 × 8192 | 13 | 142 | 9 | 4.05 | **2.18** | **97.4** |
| **FocalCodec@25** | 0.33 | 16 | 25.0 | 1 × 8192 | 13 | 144 | 9 | 4.14 | 3.30 | 96.3 |
| **FocalCodec@12.5** | 0.16 | 16 | 12.5 | 1 × 8192 | 13 | 145 | 8 | **4.22** | 7.94 | 93.9 |

## G.2 Subjective Evaluation

We conduct a subjective test with 40 participants who rate a total of 10 reconstructions from LibriSpeech `test-clean`. Following prior work [14, 86, 37, 48], we employ the MUSHRA [60] format without hidden anchor. Listeners compare multiple versions of an example at once, including a labeled reference and a hidden reference. They are asked the following question: "*Please evaluate the quality proximity between an audio sample and its reference. Please listen carefully to the reference audio and then rate the quality of each test audio clip compared to the reference. Use the scale where 0 indicates no resemblance to the reference, and 100 means perfectly the same as the reference.*" Participants were recruited online by sharing a link to the test across various public channels. To keep the subjective test short, we selected a subset of baselines based on their overall performance in objective metrics. To ensure that participants spent sufficient time on each listening task, we filtered out submissions where less than 60 seconds were spent on any of the 10 reconstructions. Out of 40 total submissions, this resulted in 33 valid entries. As showcased in Figure 2, FocalCodec achieves extremely low bitrates while maintaining strong performance. In particular, FocalCodec@50 outperforms most baselines and remains comparable to BigCodec and Stable Codec.

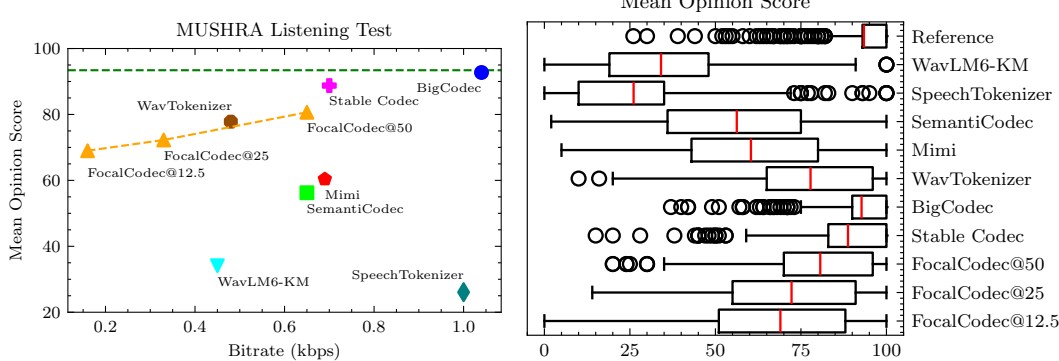

Figure 2: Subjective evaluation from 33 participants averaged over 10 samples. **Left.** Trade-off between mean opinion score and bitrate. The green dashed line highlights the reference score. Focal-Codec achieves extremely low bitrates while maintaining strong performance. **Right.** Distribution of mean opinion score. The red lines highlight the mean. FocalCodec@50 outperforms most baselines and remains comparable to BigCodec and Stable Codec.

## G.3 Streaming Inference

Although our codec is non-causal, it can be made streamable via chunked inference. This involves splitting the input signal into fixed-size chunks with a certain amount of overlap to reduce boundary artifacts. To assess the streamability of our codec, we use a chunk size of 500 milliseconds with a ~4% overlap and a left context of 3 seconds. The reconstructed chunks are stitched together using the overlap-add method with linear fade-in/fade-out. As shown in Table 8 (upper section), despite the gap with the full-context model, FocalCodec@50 maintains acceptable performance at 500 milliseconds latency. However, this is still too high for strict real-time use. To enable real-time streaming, we replace all non-causal components with their causal counterparts, and we distill the non-causal WavLM features into the causal model using a feature-matching loss during training. Additionally, we train the decoder to reconstruct 24 kHz waveforms. As shown in Table 8 (lower section), with these adjustments, together with increased model capacity and additional training data, we can achieve competitive performance at 80 milliseconds latency while maintaining a real-time factor suitable for deployment on consumer-grade GPUs.

Table 8: Streaming clean speech resynthesis on LibriSpeech `test-clean` [47].

| Codec | Bitrate (kbps) | Sample Rate (kHz) | Token Rate (Hz) | Codebooks | Causal | Latency (ms) | UTMOS ↑ | dWER ↓ | Sim ↑ |
|---|---|---|---|---|---|---|---|---|---|
| **FocalCodec@50** | 0.65 | 16 | 50.0 | 1 × 8192 | ✗ | Inf | **4.05** | **2.18** | **97.4** |
| **FocalCodec@50** | 0.65 | 16 | 50.0 | 1 × 8192 | ✗ | 500 | 3.16 | 4.55 | 96.9 |
| EnCodec | 1.50 | 24 | 75.0 | 2 × 1024 | ✓ | 20 | 1.58 | 8.08 | 93.8 |
| Mimi | 0.69 | 24 | 12.5 | 5 × 2048 | ✓ | 80 | 3.29 | 5.73 | 96.0 |
| **FocalCodec@50** | 0.60 | 24 | 50.0 | 1 × 4096 | ✓ | 80 | **3.87** | **4.38** | 96.3 |

### G.4 Mel-Spectrogram Analysis

Figure 3 shows examples of reconstructed Mel-spectrograms from LibriSpeech [47] (left) and Libri1Mix [12] (right), using the 3 top-performing codecs. The reconstructed speech from LibriSpeech is almost indistinguishable from the ground truth. For Libri1Mix, the first row shows audio contaminated with noise, while the second row shows the original clean audio. It can be observed that BigCodec, a purely acoustic codec trained for reconstruction, attempts to reconstruct the noise, resulting in poor intelligibility. In contrast, Stable Codec and FocalCodec, which have semantically meaningful representations, are able to perform basic denoising. Notably, FocalCodec assigns more energy to the frequency bands corresponding to speech, even more than in the original clean audio, leading to improved intelligibility. On the other hand, Stable Codec, while providing good denoising, introduces some artifacts and static noise in the lower part of the spectrogram, which degrades intelligibility.

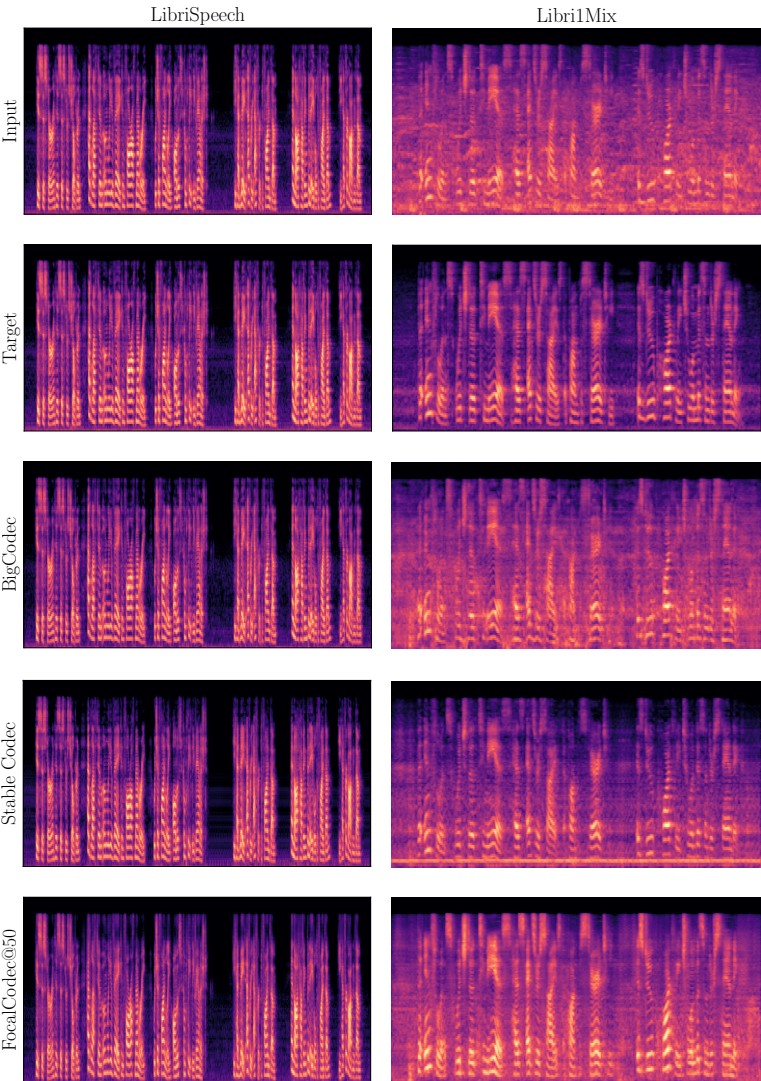

Figure 3: Reconstructed Mel-spectrograms from LibriSpeech [47] (left) and Libri1Mix [12] (right).

