# OpenReview forum: "FocalCodec: Low-Bitrate Speech Coding via Focal Modulation Networks"
_NeurIPS.cc/2025/Conference — NeurIPS 2025 poster_

### Official Review · Reviewer_p8pB · 2025-06-30

**Clarity:** 3
**Significance:** 3
**Originality:** 3
**Rating:** 5
**Confidence:** 4

**Summary:**

This paper proposed a low bit rate speech codec system called FocalCodec. Different from most of the neural audio codec which comprised of encoder, decoder and quantizer, FocalCodec added focal modulation-based Compressor and Decompressor modules. For quantizer, it used binary spherical quantization method (BSQ). Both focal modulation net and BSQ were originally used for image domain. Comprehensive experiments are conducted showing promising performance of FocalCodec in terms of both reconstruction task and down streaming tasks.

**Questions:**

Can FocalCodec be applied to real-time down stream tasks? If so, what are the supported latency?

**Ethical Concerns:**

["NO or VERY MINOR ethics concerns only"]

**Final Justification:**

This paper proposed a low bit rate speech codec system by adopting focal modulation-based Compressor and Decompressor modules. For quantization, binary spherical quantization method is used to convert continuous embeddings into discrete tokens. Comprehensive experiments have been conducted and promissing results are achieved. During rebuttal phase, the authors clarified the question on possibility of real-time system I raised as well as the weakness of insufficient comparison with other model architecture I mentioned. The clarification is reasonable and answered the questions and cleared my initial concerns. Particularly, for converting the current system into realtime system, they provided extra experiments and results, and show the prominence of real-time capability. After the final review, I stick to my initial score, which is 5-Accetp.

**Limitations:**

yes

**Quality:**

3

**Strengths And Weaknesses:**

Strengths:

1, The paper is well-written and highly readable.
2, Technique is sounding. Although initially developed for image and video compression, the proposed methods (Focal modulation net and BSQ) are creatively repurposed to offer a compelling and successful approach to speech codec design. The proposed two-stage training scheme which decouple the optimization for quantization and speech reconstruction modules is also critical for building a high-quality codec system given limited amount of model parameter.
3, Experiments are comprehensive, and performance is promising.

Weakness:

For the experiment part, to demonstrate the effectiveness of focal net, it would be necessary to show the performance comparison of focal net with convolution net and transformer architecture. Moreover, to demonstrate the advantage of the proposed BSQ quantizer, it would be also necessary to compare it with the often-used vector quantization, e.g, the one used in DAC.

---

> ### Author Rebuttal · Authors · 2025-07-31
>
> We thank the reviewer for their constructive feedback. Below we address the main points raised.
>
> **"For the experiment part, to demonstrate the effectiveness of focal net, it would be necessary to show the performance comparison of focal net with convolution net and transformer architecture. Moreover, to demonstrate the advantage of the proposed BSQ quantizer, it would be also necessary to compare it with the often-used vector quantization, e.g, the one used in DAC."**
>
> We kindly note that a similar comparison is already provided in Table 5 (Ablation Studies) of the main paper. In this table, we compare FocalNet-based quantizers against strong alternatives, including:
> - Conformer, widely regarded as an enhanced version of the vanilla Transformer and extensively used in speech modeling;
> - AMP blocks, specifically designed for speech synthesis and shown to outperform standard convolutional blocks (e.g., in BigVGAN [1]).
> We believe these are more representative and challenging baselines than standard Transformer or ConvNet architectures, and our results demonstrate a clear advantage of FocalNet in this context.
> Regarding the quantization method, the same table also includes a comparison between our Binary Spherical Quantizer (BSQ) and Finite Scalar Quantizer (FSQ). FSQ has recently been shown to outperform traditional vector quantization methods, such as the one used in DAC. Therefore, our ablation study already provides evidence of BSQ’s effectiveness over stronger and more recent baselines than standard VQ.
>
> [1] S.-G. Lee, W. Ping, B. Ginsburg, et al. “BigVGAN: A Universal Neural Vocoder with Large-Scale Training”. ICLR, 2023.
>
>
> **"Can FocalCodec be applied to real-time down stream tasks? If so, what are the supported latency?"**
>
> The current focal modulation module, comprising full-context convolutions, linear layers, and global pooling, is non-causal, as is the WavLM encoder due to its use of full-context attention. However, in practice, we find that the effective receptive field of these components is limited to just a few seconds of future context. This allows us to perform streaming inference by processing the audio in overlapping chunks of 500 ms without requiring any changes to the model architecture.
> Below are the results of chunked inference for the speech resynthesis task on LibriSpeech for FocalCodec@50:
>
> | Latency    | UTMOS ↑ | dWER ↓ | Sim ↑ |
> | ---------- | ------- | ------ | ----- |
> | Infinity   | 4.05    | 2.18   | 97.4 |
> | 500 ms     | 3.16    | 4.55   | 91.0 |
>
>
> While a 500 ms latency is acceptable for semi-streaming applications, it remains too high for real-time use. To address this, we are actively exploring architectural modifications to enable fully causal inference. These changes include:
> 1) Replacing full-context attention with chunked or causal attention;
> 2) Replacing standard convolutions with causal convolutions;
> 3) Replacing global average pooling with running mean pooling;
> 4) Distilling the non-causal WavLM features into the causal model using a feature matching loss during training.
>
> Preliminary results reported below suggest that, with these adjustments, and by scaling up model capacity and amount of training hours, we can achieve competitive performance at 80 ms latency, which is the same as Mimi, while maintaining a real-time factor high enough for deployment on consumer-grade GPUs:
>
> | Codec          | Bitrate (kbps) | Codebooks | Latency | UTMOS ↑   | dWER ↓   | Sim ↑      |
> |----------------|----------------|-----------|---------|--------- |--------  |---------  |
> | FocalCodec@50  | **0.60**           | 1×4096    | 80 ms   | **3.87** | **4.38** | **96.3** |
> | Mimi           | 0.69           | 5×2024    | 80 ms   | 3.29     | 5.73     | 96.0     |
> | EnCodec        | 1.50           | 2×1024    | 20 ms   | 1.58     | 8.08     | 93.8     |

---

> > ### Comment · Reviewer_p8pB · 2025-08-06
> >
> > Thanks for clarifying my questions and sharing the corresponding experiment results. It's encouraging to see the evaluation outcomes aligned with specific latency in the causal setting. While the performance declined under the real-time conditions compared to the batch-style processing, the trade-off is understandable given the support for real-time capabilities.

---

### Official Review · Reviewer_UCYX · 2025-07-01

**Clarity:** 4
**Significance:** 2
**Originality:** 3
**Rating:** 4
**Confidence:** 4

**Summary:**

The paper introduces FocalCodec, a novel low-bitrate neural audio codec designed to efficiently compress speech into discrete tokens using a single binary codebook. Unlike previous methods that often rely on complex multi-codebook architectures, FocalCodec employs a focal modulation-based compressor-quantizer-decompressor framework. The use of binary spherical quantization (BSQ), along with certain design details (e.g. the choice of activation function), reflects the thoroughness and novelty of the work.. This design preserves both semantic and acoustic information at bitrates as low as 0.16 kbps, establishing competitive performance with prior codecs in terms of reconstruction quality, voice conversion, and downstream task performance. The authors demonstrate that FocalCodec is effective for both discriminative tasks (like ASR and speaker ID) and generative tasks (like TTS and speech separation), offering a strong balance of compression, quality, and versatility.

**Questions:**

* Can this method be applied in streaming scenarios? It appears that the focal modulation module is non-causal. Is this structure easy to adapt for streaming use, and how would such a modification affect performance?
* FocalCodec seems to achieve good performance without relying heavily on complex training strategies. If additional techniques such as semantic distillation were introduced, could the performance be further improved?
* The use of a single codebook can indeed reduce system complexity; however, it typically requires a larger codebook, which may introduce challenges for subsequent joint modeling of text and speech. In this work, a codebook size of 8192 is used—could this size be further reduced?

**Ethical Concerns:**

["NO or VERY MINOR ethics concerns only"]

**Final Justification:**

In the rebuttal, the authors supplemented their work with human evaluations and additional experiments, confirming the feasibility of the proposed method in streaming scenarios. They also conducted a more in-depth discussion on issues such as the size of the codebook. Although the method performed slightly worse than BigCodec and Stable Codec in human evaluations, it still demonstrated competitive performance. Therefore, I will maintain my initial score and remain inclined to accept this paper.

**Limitations:**

Yes

**Quality:**

3

**Strengths And Weaknesses:**

* Strengths:
    * The idea presented in this paper is novel. Unlike previous works that primarily focus on improving performance through the design of training strategies, this paper explores enhancing speech codec performance from the perspective of model architecture and demonstrates competitive results.
    * The bitrate achieved in this work is sufficiently low while maintaining acceptable performance, which confirms the feasibility of the proposed approach.
    * The writing of the paper is clear and easy to follow.

* Weaknesses:
    * The proposed method should be validated through larger-scale experiments, using a larger training set to further demonstrate its effectiveness.
    * Subjective evaluations are missing, such as MOS or MUSHRA scores. As the authors mentioned, some objective metrics may not align well with human perception, which further highlights the necessity of including subjective assessments.

Overall, the idea in the paper is good. I would be happy to raise my score if more convincing experimental results or evaluation methods can be provided.

---

> ### Author Rebuttal · Authors · 2025-07-31
>
> We thank the reviewer for their constructive feedback. Below we address the main points raised.
>
> **"Subjective evaluations are missing, such as MOS or MUSHRA scores. As the authors mentioned, some objective metrics may not align well with human perception, which further highlights the necessity of including subjective assessments."**
>
> We conduct a subjective test with 40 participants who rate a total of 10 reconstructions from LibriSpeech test-clean. Following prior work, we employ the MUSHRA format without hidden anchor. Listeners compare multiple versions of an example at once, including a labeled reference and a hidden reference. They are asked the following question: "Please evaluate the quality proximity between an audio sample and its reference. Please listen carefully to the reference audio and then rate the quality of each test audio clip compared to the reference. Use the scale where 0 indicates no resemblance to the reference, and 100 means perfectly the same as the reference." Participants were recruited online by sharing a link to the test across various public channels. To keep the subjective test short, we did not include EnCodec and DAC due to their poor performance based on objective metrics.
> To ensure that participants spent sufficient time on each listening task, we filtered out submissions where less than 60 seconds were spent on any of the 10 reconstructions. Out of 40 total submissions, this resulted in 33 valid entries. The results reported below confirm that FocalCodec achieves extremely low bitrates while maintaining strong performance. In particular, FocalCodec@50 outperforms most baselines and remains comparable to BigCodec and Stable Codec.
>
> | Codec            | Mean  | 95% CI (Lower-Upper) |
> | ---------------- | ----- | -------------------- |
> | WavLM6-KM        | 34.10 | 31.81 – 36.38        |
> | SpeechTokenizer  | 26.08 | 23.70 – 28.46        |
> | SemantiCodec     | 56.25 | 53.61 – 58.88        |
> | Mimi             | 60.38 | 57.77 – 63.00        |
> | WavTokenizer     | 77.81 | 75.51 – 80.10        |
> | BigCodec         | 92.72 | 91.49 – 93.96        |
> | Stable Codec     | 88.76 | 87.15 – 90.36        |
> | FocalCodec@50    | 80.65 | 78.63 – 82.66        |
> | FocalCodec@25    | 72.25 | 69.92 – 74.59        |
> | FocalCodec@12_5  | 68.99 | 66.53 – 71.46        |
>
>
> **"Can this method be applied in streaming scenarios? It appears that the focal modulation module is non-causal. Is this structure easy to adapt for streaming use, and how would such a modification affect performance?"**
>
> The current focal modulation module, comprising full-context convolutions, linear layers, and global pooling, is non-causal, as is the WavLM encoder due to its use of full-context attention. However, in practice, we find that the effective receptive field of these components is limited to just a few seconds of future context. This allows us to perform streaming inference by processing the audio in overlapping chunks of 500 ms without requiring any changes to the model architecture.
> Below are the results of chunked inference for the speech resynthesis task on LibriSpeech for FocalCodec@50:
>
> | Latency    | UTMOS ↑ | dWER ↓ | Sim ↑ |
> | ---------- | ------- | ------ | ----- |
> | Infinity   | 4.05    | 2.18   | 97.4 |
> | 500 ms     | 3.16    | 4.55   | 91.0 |
>
> While a 500 ms latency is acceptable for semi-streaming applications, it remains too high for real-time use. To address this, we are actively exploring architectural modifications to enable fully causal inference. These changes include:
> 1) Replacing full-context attention with chunked or causal attention;
> 2) Replacing standard convolutions with causal convolutions;
> 3) Replacing global average pooling with running mean pooling;
> 4) Distilling the non-causal WavLM features into the causal model using a feature matching loss during training.
>
> Preliminary results reported below suggest that, with these adjustments, and by scaling up model capacity and amount of training hours, we can achieve competitive performance at 80 ms latency, which is the same as Mimi, while maintaining a real-time factor high enough for deployment on consumer-grade GPUs:
>
> | Codec          | Bitrate (kbps) | Codebooks | Latency | UTMOS ↑   | dWER ↓   | Sim ↑      |
> |----------------|----------------|-----------|---------|--------- |--------  |---------  |
> | FocalCodec@50  | **0.60**           | 1×4096    | 80 ms   | **3.87** | **4.38** | **96.3** |
> | Mimi           | 0.69           | 5×2024    | 80 ms   | 3.29     | 5.73     | 96.0     |
> | EnCodec        | 1.50           | 2×1024    | 20 ms   | 1.58     | 8.08     | 93.8     |
>
>
> **"The proposed method should be validated through larger-scale experiments, using a larger training set to further demonstrate its effectiveness."**
>
> As mentioned in our previous point, for the causal variant experiment, we used a larger training dataset to compensate for the performance drop due to switching from full-context to low-latency settings. Specifically, we experimented at two scales: the medium subset of LibriLight, consisting of approximately 5k hours of English speech, and the full LibriLight dataset, comprising around 60k hours. The table below shows how scaling up the training data improves performance, consistent with the scaling behavior often seen in deep learning.
>
> | Training Hours | UTMOS ↑ | dWER ↓ | Sim ↑ |
> | -------------- | ------- | ------ | ----- |
> | 5k             | 3.78    | 4.98   | 95.7  |
> | 60k            | **3.87**| **4.38**   | **96.3**  |
>
>
> **"FocalCodec seems to achieve good performance without relying heavily on complex training strategies. If additional techniques such as semantic distillation were introduced, could the performance be further improved?"**
>
> Since FocalCodec relies on a single shared codebook, it is important to be cautious about the type of distillation applied. For example, semantic distillation (as in Mimi) could enhance semantic content but may degrade acoustic detail due to the shared representation space. In contrast, more balanced distillation strategies that combine both semantic and acoustic supervision (such as jointly optimizing ASR and speaker identification objectives) may help improve both aspects without compromising either. That said, as the reviewer rightly noted, one of the key strengths of our approach is its simplicity. A more straightforward and robust path to improved performance may be to retrain WavLM with increased capacity or on a larger and more diverse dataset. This would likely enhance the quality of the representations while preserving the balance between semantic and acoustic information.
>
>
> **"The use of a single codebook can indeed reduce system complexity; however, it typically requires a larger codebook, which may introduce challenges for subsequent joint modeling of text and speech. In this work, a codebook size of 8192 is used, could this size be further reduced?"**
>
> While 8192 tokens may seem large, it is modest compared to vocabulary sizes commonly used in modern text LLMs -- for example, LLaMA 3 uses 128k tokens, and Gemma 2 reaches up to 256k. Even within the speech domain, recent codecs like TS3-Codec use vocabularies exceeding 100k tokens.
> To investigate the impact of reducing the codebook size, we conducted an experiment using a codebook size of 1024 instead of 8192. Results for FocalCodec@50 on LibriSpeech resynthesis are shown below:
>
> | Codebook Size | UTMOS ↑ | dWER ↓ | Sim ↑ |
> | ------------- | ------- | ------ | ----- |
> | 8192          | 4.05    | **2.18**   | **97.4**  |
> | 1024          | **4.14**    | 2.54   | 95.3  |
>
> As expected, reducing the codebook size primarily impacts speaker similarity, since larger codebooks are better at capturing fine-grained acoustic details such as speaker identity. Notably, this experiment was conducted without increasing model capacity or training data. Therefore, it is reasonable to expect that performance with as few as 1024 codes could be further improved by scaling up the model and training on larger datasets.

---

> > ### Comment · Reviewer_UCYX · 2025-08-06
> > **follow-up**
> >
> > I would like to thank the authors for their detailed response, which has alleviated some of my key concerns.

---

### Official Review · Reviewer_mS6Q · 2025-07-05

**Clarity:** 4
**Significance:** 3
**Originality:** 3
**Rating:** 4
**Confidence:** 4

**Summary:**

This paper proposes FocalCodec, a novel single-codebook hybrid speech codec based on focal modulation networks and binary spherical quantization (BSQ). It achieves low bitrates (0.16–0.65 kbps) while preserving both semantic and acoustic information, outperforming prior work in reconstruction tasks and downstream ASR/SI/SER/TTS tasks. Its architecture combines a frozen WavLM encoder, compressor with focal modulation, BSQ quantizer, decompressor, and an efficient Vocos-based decoder. Extensive experiments demonstrate the performance for low-bitrate speech coding.

**Questions:**

1. The performance of DAC in this paper is much worse than the original paper. Did you use and evaluate the DAC properly?
2. The performance on downstream tasks seems limited. The WER for TTS task is around 30, it has room to improve.
3. This paper use pretrained WavLM feature as input. This limits the generalization ability for codec, which is very important for codec feature.

**Ethical Concerns:**

["NO or VERY MINOR ethics concerns only"]

**Final Justification:**

I have carefully considered the authors’ rebuttal and appreciate the detailed clarifications and additional experiments they provided. While some concerns remain, the response successfully addressed several critical points I raised. My final decision is to raise my initial score to 4. This paper explores the potential of low bit codec in the downstream tasks, and the authors shows reasonable results on TTS in the rebuttal response.

**Limitations:**

As mentioned in last session, the author’s results is not aligned with other previous paper. Besides, the downstream tasks’ generated samples are not showed on the demo page. So I highly recommend the author to further check the experimental part and improve downstream tasks’ performance. More details please refer to the Questions part.

**Paper Formatting Concerns:**

After reviewing the paper, no major formatting violations were observed

**Quality:**

3

**Strengths And Weaknesses:**

Strengths:
1. Ultra-low bitrate performance: This work achieves 0.16 kbps while preserving intelligibility and speaker similarity, surpassing many baselines operating at >1 kbps.
2. This codec using only one codebook, encoding both semantic information and acoustic information, which is suitable for both recognition and generative downstream tasks.
3. This paper contains both code and demo page, shows good reproducibility.
Weaknesses:
1. DAC baseline evaluation concerns. The reported DAC performance is much worse than in its original paper.
2. Limited downstream task performance. For example, TTS WER remains around 30, indicating poor intelligibility.
3. Pretrained WavLM encoder dependency. Using frozen WavLM features as input limits generalization, as it relies heavily on the pretraining domain of WavLM.

---

> ### Author Rebuttal · Authors · 2025-07-31
>
> We thank the reviewer for their constructive feedback. Below we address the main points raised.
>
> **"DAC baseline evaluation concerns. The reported DAC performance is much worse than in its original paper."**
>
> In our experiments, we evaluate DAC in a low-bitrate setting (1 kbps) by using only 2 out of its 32 available codebooks, which significantly degrades performance compared to the 32-codebook variant. Additionally, we use the 16 kHz version of DAC, which has a lower bitrate but underperforms relative to the 24 kHz version used in the original DAC paper. When we evaluate the 24 kHz version using our resynthesis pipeline on LibriSpeech, we obtain the following results:
>
> | Codec | Bitrate (kbps) ↓ | UTMOS ↑ | dWER ↓ | Sim ↑ | Code Usage ↑ | Norm. Entropy ↑ |
> | --------- | ------- | ----- | ----- | ---- | ---------- | ------------- |
> | DAC-24kHz | 1.50    | 1.79  | 12.67 | 92.1 | 99.9       | 86.6          |
> | DAC-16kHz | 1.00    | 1.29  | 20.04 | 89.2 | 100.0      | 91.7          |
>
> These results are better than the 16 kHz version but still significantly behind other baselines, despite the higher bitrate (1.5 kbps of DAC-24kHz vs 1 kbps of DAC-16kHz).
> We hope this explanation clarifies the observed performance gap and helps convince the reviewer of the reliability of our evaluation. We clarified this aspect in the paper.
>
>
> **"Limited downstream task performance. For example, TTS WER remains around 30, indicating poor intelligibility."**
>
> We agree that there is room for improvement in TTS performance. However, it's important to note that our TTS models are trained on only 460 hours of speech -- several orders of magnitude less than what is typically used in large-scale TTS systems. Our goal is to compare codec representations fairly in a controlled setting, rather than to achieve state-of-the-art speech synthesis.
> That said, we considered the reviewer's suggestion and explored ways to improve TTS quality.
> First, we observed that LibriSpeech-460 contains very few utterances longer than 20 seconds, whereas LibriSpeech test-clean includes several such samples (around 4%). To reduce mismatch between training and testing conditions, we removed these long utterances from the test set.
> Second, following the same experimental protocol from [1], we generated multiple samples (using temperature = 1.0 and top-p = 0.9) and selected the one with the lowest WER relative to the input text. By sampling 5 outputs per input and choosing the best one (using Whisper-small for transcription) we observed improved intelligibility across all TTS models.
> Below are the updated results, which further emphasize the effectiveness of our codecs.
>
> | Codec            | UTMOS ↑  | dWER ↓      | Sim ↑     | Mel Distance ↓ |
> | ---------------- | -------- | --------- | -------- | -------------- |
> | EnCodec          | 1.71     | 64.28     | 83.2     | 131.00         |
> | DAC              | 1.34     | 47.06     | 85.9     | 113.00         |
> | WavLM6-KM        | 3.74     | 38.67     | 88.7     | 98.85          |
> | SpeechTokenizer  | 2.69     | 35.46     | 89.2     | 93.51          |
> | SemantiCodec     | 2.82     | 48.38     | 91.4     | 99.47          |
> | Mimi             | 3.11     | 28.63     | **93.6** | 101.00         |
> | WavTokenizer     | 3.68     | 47.56     | 92.8     | 94.52          |
> | BigCodec         | 3.43     | 54.43     | 89.4     | 105.00         |
> | Stable Codec     | 3.19        | 49.28         | 88.8        | 101              |
> | FocalCodec@50    | 4.11     | 28.10     | 93.3     | 90.86          |
> | FocalCodec@25    | **4.16** | **16.75** | 91.6     | 88.01          |
> | FocalCodec@12_5  | 4.12     | 21.59     | 90.8     | **86.14**      |
>
> [1] J. Tian, J. Shi, W. Chen, et al. “ESPnet-SpeechLM: An Open Speech Language Model Toolkit”. NAACL, 2025.
>
>
> **"Pretrained WavLM encoder dependency. Using frozen WavLM features as input limits generalization, as it relies heavily on the pretraining domain of WavLM."**
>
> We would like to note that in our opinion domain limitations apply to all models including ours. For instance, EnCodec performs well on clean speech but struggles in noisy conditions, as it was not trained on such data. Similarly, our method inherits the limitations of the WavLM encoder's pretraining domain, but this is a general characteristic of pretrained models.
> That said, our quantization framework is not tied to WavLM. We use WavLM in this work due to its strong performance in speech modeling, but the compressor-quantizer-decompressor pipeline is modular and can, in principle, be applied to any encoder. For example, it could be fine-tuned on top of a continuous autoencoder trained for general audio or music.

---

> > ### Comment · Reviewer_mS6Q · 2025-08-05
> > **Thanks for the rebuttal and I will raise my rating to 4.**
> >
> > I have carefully considered the authors’ rebuttal and appreciate the detailed clarifications and additional experiments they provided. While some concerns remain, the response successfully addressed several critical points I raised. My final decision is to raise my initial score to 4. This paper explores the potential of low bit codec in the downstream tasks, and the authors shows reasonable results on TTS in the rebuttal response.

---

### Official Review · Reviewer_Qfbg · 2025-07-06

**Clarity:** 3
**Significance:** 4
**Originality:** 3
**Rating:** 4
**Confidence:** 5

**Summary:**

This paper proposes FocalCodec, a low-bitrate speech codec based on focal modulation and a single binary codebook. It aims to retain both acoustic and semantic information while simplifying the architecture compared to multi-codebook approaches. The method shows competitive performance on speech resynthesis, voice conversion, and various downstream tasks, particularly at bitrates as low as 0.16 kbps.

**Questions:**

1. On the explanation of Table 4 (refer to weaknesses).
2. In training stage 2, the continuous representations before quantization are used as input to the decoder. Could you provide more details on this process? Alternatively, would it be possible to include a "skip quantization" option in the code to facilitate testing?
3. We noticed that low-bitrate codecs in Table 4 generally yield high WER in generative tasks such as TTS. Although your model is trained on less than 500 hours of data, training a higher-bitrate codec (e.g., EnCodec or DAC) or a continuous-representation-based TTS model on the same amount of data would likely yield much better results. What do you think is the main reason for this performance gap? Does it suggest that low-bitrate codecs are inherently unsuitable for generative tasks?

**Ethical Concerns:**

["NO or VERY MINOR ethics concerns only"]

**Final Justification:**

I appreciate the additional results provided, which clarified my concerns regarding the main experimental results. I will maintain my positive score.

**Limitations:**

Yes

**Quality:**

4

**Strengths And Weaknesses:**

**Strengths:**

1. The paper is well-written and clearly organized.
2. The work is technically sound, with comprehensive experiments. It demonstrates strong performance even at extremely low frame rates, including on multilingual and noisy speech reconstruction. It also performs well on voice conversion and is thoroughly evaluated on both discriminative (e.g., ASR) and generative (e.g., TTS) downstream tasks.
3. The model design and training process are novel in several aspects, such as the use of focal downsampling and BSQ, the parallel two-stage training strategy, and data augmentation to improve noise robustness.

**Weaknesses:**

1. The newly introduced modules, such as the focal module and BVQ, are all based on previous work. Although the final experimental results are good, the work lacks fundamental innovation.
2. The introduction to the focal modulation in Section 3.1 is not intuitive enough and could be illustrated with diagrams for better understanding.
3. In Table 4, results on generative tasks show that FocalCodec can yield high dWER while achieving DNSMOS/UTMOS scores close to or even better than the reference. This discrepancy is confusing. It would be helpful to include additional metrics (e.g., MCD for TTS) for a more balanced evaluation.

---

> ### Author Rebuttal · Authors · 2025-07-31
>
> We thank the reviewer for their constructive feedback. Below we address the main points raised.
>
> **"The newly introduced modules, such as the focal module and BVQ, are all based on previous work. Although the final experimental results are good, the work lacks fundamental innovation."**
>
> While the focal modulation module and binary spherical quantization build upon prior ideas, the key innovation lies in how they are integrated into a simple and effective framework. A key contribution, overlooked in prior work, is that quantizing a pretrained self-supervised model can directly yield tokens that are both acoustic and semantic. This challenges the common belief that one must first train an acoustic autoencoder and then inject semantics via distillation or other complex techniques. Instead, our approach shows that a single quantizer operating on low-level self-supervised features can achieve both goals simultaneously within a low-bitrate, efficient framework. While simple in hindsight, we believe this represents a novel and impactful insight as evidenced by experimental results.
>
>
>
> **"The introduction to the focal modulation in Section 3.1 is not intuitive enough and could be illustrated with diagrams for better understanding."**
>
> We agree that the explanation in Section 3.1 can be made more intuitive to improve clarity. We will revise the paragraph accordingly in the updated version of the paper.
>
>
> **"In Table 4, results on generative tasks show that FocalCodec can yield high dWER while achieving DNSMOS/UTMOS scores close to or even better than the reference. This discrepancy is confusing. It would be helpful to include additional metrics (e.g., MCD for TTS) for a more balanced evaluation."**
>
> UTMOS measures naturalness of the speech signal, not intelligibility or semantic accuracy. It is entirely possible for speech to sound fluent and pleasant (high UTMOS) even if the content is incorrect or semantically invalid (high dWER). Likewise, UTMOS does not reflect whether the speaker identity is preserved. This is why it is important to report complementary metrics to capture different aspects of generative quality (UTMOS <-> naturalness, dWER <-> intelligibility, speaker similarity <-> speaker identity preservation).
> We agree that including an additional metric is helpful. In fact, in our updated TTS experiments (see the response to Reviewer mS6Q for more details), we now also report Mel Distance to provide a more comprehensive evaluation. Even with the addition of this metric, the results remain consistent with our previous findings.
>
> | Codec            | UTMOS ↑  | dWER ↓      | Sim ↑     | Mel Distance ↓ |
> | ---------------- | -------- | --------- | -------- | -------------- |
> | EnCodec          | 1.71     | 64.28     | 83.2     | 131.00         |
> | DAC              | 1.34     | 47.06     | 85.9     | 113.00         |
> | WavLM6-KM        | 3.74     | 38.67     | 88.7     | 98.85          |
> | SpeechTokenizer  | 2.69     | 35.46     | 89.2     | 93.51          |
> | SemantiCodec     | 2.82     | 48.38     | 91.4     | 99.47          |
> | Mimi             | 3.11     | 28.63     | **93.6** | 101.00         |
> | WavTokenizer     | 3.68     | 47.56     | 92.8     | 94.52          |
> | BigCodec         | 3.43     | 54.43     | 89.4     | 105.00         |
> | Stable Codec     | 3.19        | 49.28         | 88.8        | 101              |
> | FocalCodec@50    | 4.11     | 28.10     | 93.3     | 90.86          |
> | FocalCodec@25    | **4.16** | **16.75** | 91.6     | 88.01          |
> | FocalCodec@12_5  | 4.12     | 21.59     | 90.8     | **86.14**      |
>
>
> **"In training stage 2, the continuous representations before quantization are used as input to the decoder. Could you provide more details on this process? Alternatively, would it be possible to include a ‘skip quantization’ option in the code to facilitate testing?"**
>
> Yes, this is already supported. Here's a more detailed explanation:
> - During training, the decoder learns to map continuous WavLM layer-6 features to the corresponding waveform.
> - At inference time, these continuous features are passed through the compressor, quantizer, and decompressor to obtain dequantized features, which are then fed to the decoder. The decompressor is trained to reconstruct the original continuous features from the discrete codes, so the dequantized features closely approximate the originals. As a result, the decoder maintains strong performance even when using dequantized inputs, without requiring any additional fine-tuning.
>
> This means one can bypass the quantization step entirely and feed the continuous features directly into the decoder. In this configuration, the system effectively runs as a continuous autoencoder, which typically yields higher reconstruction quality than the fully quantized version. We added a sentence in Sec. 3.2 to clarify this aspect.
>
>
> **"We noticed that low-bitrate codecs in Table 4 generally yield high WER in generative tasks such as TTS... Does it suggest that low-bitrate codecs are inherently unsuitable for generative tasks?"**
>
> Table 4 suggests the opposite: low-bitrate codecs, when reconstruction quality is sufficiently high, are particularly well-suited for generative tasks, especially in autoregressive settings. In fact, our lowest-bitrate codecs achieve the best performance on TTS. This is largely due to the compactness of the token sequences they produce, which significantly simplifies sequence modeling.
>
> In contrast, higher-bitrate codecs such as DAC and EnCodec generate much longer sequences, which can pose challenges for autoregressive models, especially when training data is limited. Notably, not only the WER but also other metrics deteriorate for higher-bitrate codecs - for example, EnCodec shows worse results across the board despite its higher bitrate.
> More broadly, generative modeling over discrete speech tokens involves a fundamental trade-off:
> - High bitrate -> easier to achieve high reconstruction fidelity, but results in long sequences that are difficult to model;
> - Low bitrate -> harder to preserve fine-grained details, but produces shorter sequences that are more tractable for generative models.
> Our results show that a single, low-bitrate codebook can strike an effective balance between reconstruction quality and generative tractability.

---

> > ### Comment · Reviewer_Qfbg · 2025-08-06
> >
> > Thank you for the response. I now have a better understanding of the novelty of the paper. I also appreciate the additional results provided, which clarified my concerns regarding the main experimental results. I will maintain my positive score.

---

### Comment · Area_Chair_rzwP · 2025-08-04
**rebuttal acknowledgement and discussion**

Dear reviewers,

Please make sure you acknowledge that you have read the authors' rebuttal, which is mandatory this year.  Also, please discuss with the authors if you have questions or comments on their rebuttal.

Thanks,

AC

---

### Decision · Program_Chairs · 2025-09-17

**Decision:**

Accept (poster)

**Comment:**

This paper proposes FocalCodec, a low-bitrate audio neural codec based on focal modulation. Compared to the existing audio neural codecs, FocalCodec achieves low bitrates and only needs a single binary codebook to compress speech between 0.16 and 0.65 kbps. It is shown to be efficiently preserve semantic and acoustic information. The authors conduct experiments to demonstrate its superior performance in speech resynthesis and voice conversion compared to numerous existing approaches. Overall, this is an interesting work that is technically solid. Concerns raised by the reviewers include questions on novelty, generalization capability and interpretation of some experimental results. The authors have cleared most of the concerns in their rebuttal. Overall, the proposed FocalCodec is sufficiently novel. The paper is well written and easy to follow. Experiments are controlled and yet extensive, which include the experimental results reported in the submission and added later on in the rebuttal. All reviewers are supportive of accepting this paper.